# Utilizing High-Resolution Mass Spectrometry Data Mining Strategy in R Programming Language for Rapid Annotation of Absorbed Prototypes and Metabolites of Gypenosides

**DOI:** 10.3390/molecules30040779

**Published:** 2025-02-07

**Authors:** Xiaoshan Li, Qianru Zhang, Yuqin Li, Lin Qin, Di Wu, Daopeng Tan, Jian Xie, Jiajia Wu, Qingping Yang, Yanliu Lu, Yongxia Zhao, Qingjie Fan, Xingdong Wu, Yuqi He

**Affiliations:** 1Guizhou Engineering Research Center of Industrial Key-Technology for Dendrobium Nobile, Zunyi Medical University, Zunyi 563009, China; li_roseixx@163.com (X.L.); zhangqianru@zmu.edu.cn (Q.Z.); lyq1286292@163.com (Y.L.); qinlin1115@163.com (L.Q.); wd_32677@126.com (D.W.); tandp@zmu.edu.cn (D.T.); xiejian@zmu.edu.cn (J.X.); yqp0513@163.com (Q.Y.); x.y.z.100@163.com (Y.Z.); fanqj@zmu.edu.cn (Q.F.); 2Shanghai Key Laboratory for Molecular Engineering of Chiral Drugs, School of Pharmacy, Shanghai Jiao Tong University, Shanghai 200240, China; wujiajia_1160@126.com; 3Key Lab of the Basic Pharmacology of The Ministry of Education, Zunyi Medical University, Zunyi 563009, China; yanliu.lu@foxmail.com

**Keywords:** R programming language, virtual metabolite library, mass defect, gypenosides, metabolite annotation

## Abstract

The rapid and accurate annotation of the complex compounds and metabolites in natural products remains a significant challenge. In this study, we developed an integrated strategy to efficiently and accurately profile both the prototypes and metabolites of natural products in vivo. And this was achieved by establishing a gypenosides constituent database and utilizing R programming language to combine sample selection, virtual metabolite database construction, polygon mass defect filtering, and Kendrick mass defect filtering. In addition, the annotation strategy was successfully applied to identify the prototypes and metabolites of gypenosides in mice serum. As a case study, gypenoside LXXV was used to validate the feasibility of this approach. The results demonstrated 36 prototypes and 108 metabolites were annotated from the serum by the established annotation strategy. The prototype and eight metabolites of gypenoside LXXV were further confirmed, indicating that the proposed strategy is available. This study provides a novel approach for the rapid and accurate identification of prototypes and metabolites of natural products and offers new insights into the metabolic processes of gypenosides in vivo.

## 1. Introduction

*Gynostemma pentaphyllum* (Thunb.) Makino (GP), also called “Jiao-Gu-Lan”, belongs to the family Cucurbitaceae and genus *Gynostemma Bl*., and is widely distributed in China, Japan, Korea, Southeast Asia and other regions [1]. In 1406 AD, it was first recorded that GP could be eaten as vegetable, and now it is widely used as a health supplement, in the form of a beverage, biscuit, noodles, face wash and bath oil [2]. GP mainly contains various saponins [3], flavonoids [4], amino acids [5], polysaccharides [6] and other components [7], which exhibit a variety of pharmacological properties, including glucose and lipid metabolism regulatory [8,9], anti-inflammatory [10], antioxidant [11], anti-tumor [12], neuroprotective [13] and anti-anxiety properties [14,15]. Modern research has shown that saponins or sapogenins are the main active components responsible for GP’s pharmacological effects [16]. For example, gypenosides (GPs) have been shown to reduce plasma AST, ALT and TG levels in mice with non-alcoholic fatty liver disease, and to alleviate hepatic steatosis induced by a high-fat, high-cholesterol diet by downregulating ACC1, PPAR, CD36, APOC3 and MTTP levels [17]. However, current research on GPs has mainly focused on their pharmacodynamics and pharmacological mechanisms. Consequently, further in-depth studies are required to ascertain whether additional saponins or sapogenins play a role in the pharmacological effects. Hence, the absorbed components characterization of GPs provides a scientific foundation for the discovery of further active components.

The chemical components in natural products undergo phase I or phase II metabolic reactions under the action of enzymes in the body, but their basic skeleton remains unchanged. Therefore, their metabolites share similar characteristics with the structure and fragmentation of the prototype compounds [18]. However, the complexity of the components in natural products, along with the occurrence of one or multiple metabolic reactions in vivo, often results in low concentrations of both the parent compounds and their metabolites, making systematic characterization challenging using traditional methods [19]. LC-MS has become an effective means for chemical composition analysis in natural products due to its multiple scan modes, high resolution and sensitivity [20,21]. However, with the development of mass spectrometry technology, high-resolution mass spectrometry (HRMS) generates vast amounts of data, most of which correspond to unknown metabolites. This poses a significant challenge in identifying target compounds from the data. Recently, an analytical technique based on high resolution mass spectrometry data, mass defect filter, has been developed. It takes advantage of the fact that analogues with the same parent structure, but different substituent groups do not have a large difference in mass loss compared with the parent compound. The mass defect filter can be employed to eliminate all interfering ions that fall outside the desired range, thus facilitating rapid screening of potential components [22]. In addition, after normalizing the *m/z* value of CH_2_ to 14 (the mass of ^12^C^1^H_2_), the researchers found that homologues (compounds differ only in the number of CH_2_ groups) have the same mass defect (MD) within 5 ppm [23]. This finding has significant implications for annotate drug metabolites if we treat the prototype component entering the body as the compound skeleton and the metabolite as a homologue of the prototype component [24,25].

Although the aforementioned techniques can be used for the rapid screening and characterization of target metabolites in complex systems, with the advancement of mass spectrometry instruments, the volume of acquired mass spectral data increases exponentially. These data often include instrumental noise, spectra generated by impurities in the solvent, and spectra of non-target components, such as adduct ion signals formed between the target component and the solvent [26]. As a result, filtering target mass spectra by manually editing formulas in Excel becomes exceedingly difficult. In recent years, computer programming languages with high-capacity memory, such as R [27], Python [28] and C [29], have been successfully applied to mass spectrometry data processing. Among them, R-based algorithms allow for offline, rapid batch searching of mass spectrometry data for dozens or even hundreds of compounds, including molecular formula, precise molecular weight, and parent ion and daughter ion information under different experimental environments. This significantly reduces the time spent by researchers in online, one-by-one searches [30].

Previously, our research group proposed a concept of using the R programming language to screen and annotate potential prototype components and metabolites in the mice serum administrated with *Dendrobium nobile* extract. However, challenges remained, particularly regarding sample selection and accurate metabolite identification [31]. In this study, based on the established gypenosides component library, an intelligent strategy was developed by integrating R programming language, sample selection, virtual metabolite library, polygon mass defect filtering and Kendrick mass defect filtering for the rapid and accurate characterization of metabolites in natural products. Additionally, this annotation strategy was applied to identify the prototypic components and metabolites in mice serum administrated with GPs. Finally, the integrated strategy was validated using gypenoside LXXV (Gyp LXXV) as an example. This approach provides a reference for identifying prototypic components and metabolites of natural products in vivo, thus supporting the discovery of more active compounds in GPs.

## 2. Results and Discussion

### 2.1. Identification of GPs Prototypes

The mass spectrum data of GPs reference standard and GPs were analyzed by UPLC-Q-TOF-MS/MS. 72 chemical components were identified from GPs, with 6 components confirmed by reference standard (Appendix A).

To obtain higher-quality mass spectrometry data of serum samples after GPs administration in mice, this study firstly compared the total ion current chromatograms (TIC) of the serum in positive and negative ion modes. The spectral responses in negative ion mode were significantly better than in positive ion mode, allowing for the detection of more serum exogenous constituents (Appendix A). Further analysis of the serum mass spectrometry data revealed the time points at which more exogenous components could be detected (Appendix A). It was observed that more exogenous components could be detected in the serum samples collected at 10, 30 and 45 min, as well as at 1, 8 and 12 h, particularly at 30 min. This suggests that most of the absorbed prototypic constituents or metabolites of GPs could be detected at these time points. However, only a small number of components were detected at 4 h and 6 h, indicating that the constituents from GPs had been mostly eliminated at these time points. These results are consistent with a pharmacokinetics study of GPs, which suggested that the time to peak of macromolecular and small-molecule GP saponins ranged from 7.79 to 12.72 h, and from 2.35 to 6.67 h, respectively [32]. In addition, compared to 4 h and 6 h, the number of detected components at 8 h and 12 h showed an increasing trend, suggesting that absorbed GPs components may be returned to the liver through enterohepatic circulation. To maximize the identification of prototypes and metabolites, serum samples at 10, 30, 45 min and 1, 8, 12 h were pooled and analyzed by UPLC-Q-TOF-MS/MS. The total ion chromatograms of blank serum and drug-containing serum are shown in Appendix A.

To reduce the large amount of redundant data generated by high resolution mass spectrometry, the polygonal mass defect filtering region was established based on the self-built virtual metabolite library to screen the absorbed prototypic components and metabolites of GPs (Figure 1A). Serum samples were analyzed in data-independent acquisition mode, and raw mass spectrometry data were subjected to deconvolution, peak alignment and peak extraction using SCIEX OS (version 3.1.6.44). As a result, 2156 mass spectrometry data were acquired in negative ion mode. After subtracting the blank background, 719 mass spectrometry data were screened out by polygon mass defect filtering, which could be preliminarily identified as prototypic components and metabolites of GPs for further identification, removing about 66% of interfering ions (Figure 1B,C).

Subsequently, blank serum and GPs were used as negative and positive control, respectively. The prototypical GPs components absorbed into the body were identified by comparing the retention time, exact molecular mass, primary and secondary fragments from the mass spectrometry data with the standards, a self-established chemical compound library, and the previously identified GPs components that appeared in both administered serum and GPs but not in blank serum. The characterization process of representative components was as follows: Compound **1**, in negative ion mode, exhibited precursor ions at *m/z* 883.50 Da and adduct ion peaks at *m/z* 929.50 Da for [M+HCOOH-H]^−^ ion. These precursor ions corresponded to a molecular formula of C_46_H_76_O_16_ with an error value of 1.7 ppm. In the MS^2^ spectrum, fragment ions were detected at *m/z* 751.46, *m/z* 605.40 and *m/z* 473.36 Da. The fragment ions at *m/z* 751.46 Da primarily resulted from losses of one molecule of xyl from the precursor ion, while the fragment ions at *m/z* 605.40 Da were attributed to the precursor ions consecutive losing a xyl and rha. Therefore, an aglycone fragment peak at *m/z* 473.36 Da was obtained through a series of deglycosylation processes. After comparing retention times and mass fragmentation with reference standards, compound **1** was identified as gylongiposide I. The MS and MS^2^ spectra, along with potential fragmentation patterns for gylongiposide I, were shown in Figure 2A. Compound **14** in the MS spectrum exhibited [M-H]^−^ ion peaks at *m/z* 897.48 Da and [M+HCOOH-H]^−^ ion peaks at *m/z* 943.48 Da. In the MS^2^ spectrum, the fragment ion at *m/z* 765.44 Da was generated by losses of one molecule of xyl from the parent ion. While the fragment ions at *m/z* 681.36 Da were attributed to the precursor ion losing an epoxy (C_5_H_8_O) in the C-23 position. The fragment ions at *m/z* 535.33 and *m/z* 473.36 Da were derived from the precursor ion by losing one molecule of rha and ara, successively. After comparing retention times and mass fragmentation with reference standards, compound **14** was identified as gypenoside A. The MS and MS^2^ spectra, along with potential frag mentation patterns for gypenoside A, were presented in Figure 2B. In conclusion, by using a combination of reference standards and literature mass data, a total of 36 prototype components of GPs were identified in mice serum. The extracting ion chromatograms (XIC) were shown as Appendix A, and the detailed information was provided in Appendix A.

### 2.2. Characterization of GPs Metabolites

The chemical composition of natural products is complex and varied, with components present in trace amounts. After entering the body, these components undergo various metabolic transformations in the blood and tissues, resulting in the formation of numerous metabolites [33,34]. Consequently, the metabolites are found at extremely low concentrations, making their identification challenging. The prototype components and metabolites of GPs in the mouse body were rapidly screened using polygon mass defect and self-built virtual metabolite library. However, the rapid identification of unknown components and metabolic types remains a challenge. Since the mass deficit (MD) theory was first proposed in 1963 [23], the further developed Kendric mass deficit (KMD) can be used to rapidly characterize homologues in mass spectrometry [35]. All homologues with the same KMD appear as a horizontal line in the plot of nominal Kendrick mass (NKM) (*X*–axis) against KMD (*Y*–axis), and the mass-to-charge ratios (*m/z*) of metabolites within the same composition differ only along the *X*–axis. Therefore, the method can be used to quickly identify specific metabolite types with known components. In this study, the method was applied to the identification of serum oxygenation and methylation metabolites after the administration of GPs in mice (Figure 3). The peak list data used in Figure 3 is provided in Appendix A.

Finally, 108 GPs metabolites were quickly screened and characterized (Appendix A). Most of these metabolites were phase I metabolites of gypenogenin, including hydroxylation, methylation and alcohol dehydration products. The results of metabolite analysis suggest that GPs may exert their effects through metabolites in the gastrointestinal tract, which is consistent with previous reports [36]. Furthermore, these findings imply that GPs may be absorbed after hydrolysis in the stomach and intestines, followed by metabolic reactions in the body. In addition to the phase I metabolic found in mice serum, phase II metabolites such as acetylation, glucuronic acid binding and sulfate binding of GPs or gypenogenin were also identified. This result aligns with the research conducted by our team, which showed that gypenogenin is metabolized by glucose-aldehyde acidification in human liver microsomal enzymes to produce new products [37]. As an example, the detected gypenoside XLIX (Gyp XLIX) in mice serum was selected for further elucidation of its metabolites and metabolic reaction in vivo (Figure 4). Gylongiposide I was generated by hydrolysis of one glucose at the C-21 position of Gyp XLIX. It is in line with the previous result that Gyp XLIX was transformed into gylongiposide I by the enzymatic biotransformation via the hydrolysis of the glucose group at the C-21 position [38]. Gylongiposide I may be transformed into M54 and M15 by occurring further oxidation and epoxidation, respectively. In addition, M56 was generated by desaturation on the parent nuclear of gylongiposide I. The metabolite M33 may be obtained by hydroxylation of its methylation product M18 or by further methylation of its hydroxylation product M43.

### 2.3. Application of Rapid Identification Methods for Natural Product Metabolites

Gyp LXXV, a major active component of GPs, was selected as an example to validate the proposed method. To obtain as many metabolites of Gyp LXXV as possible, mice were administered with Gyp LXXV (100 mg/kg) by gavage, and serum mass spectrometry data were collected at 10, 30 and 45 min, and 1, 8 and 12 h. The metabolites were quickly screened and characterized using the integrated strategy to verify the feasibility of the proposed method. As a result, the prototype of Gyp LXXV was detected in mice serum containing Gyp LXXV. Furthermore, this study indicated that Gyp LXXV can be metabolized to Compound K (*m/z* 621.43 Da) by hydrolyzing and removing a glycosyl group at C-20. In addition, the metabolites of Compound K were further found in both GPs-administrated and Gyp LXXV-administrated serum, including desaturation (M105), oxidation (M108), dealkylation (M107) and glucuronidation (M106) product. However, the metabolites of desaturation and dehydration after dealkylation were found only in the Gyp LXXV-treated serum. This may be due to the mutual interference of various components in the GPs, so that only a small amount of Gyp LXXV is absorbed, resulting in the content of its metabolites being insufficient for further metabolism. According to the metabolites after the administration of Gyp LXXV monomer, the possible metabolic pathways of Gyp LXXV in vivo are shown in Figure 5.

## 3. Materials and Methods

### 3.1. Chemicals and Materials

Gypenosides (purity ≥ 98%) were purchased from Shanxi Zhongxin Bio-Technology Co., Ltd., (Yuncheng, China). The reference standards of GPs, Gypenoside A (DST210317-013), Gypenoside XLIX (DST220519-012), Gylongiposide I (DST241023-398), Gypenoside LXXV (DST211020-248), Gypenoside LI (DST211217-280), Gypenoside XVII (DSTDS013603), Ginsenoside Rd (DSTDR001502) and Gypenoside XLVI (DST190921-111)(purity ≥ 98% by HPLC) were purchased from Chengdu Despite Biotechnology Co., Ltd., (Chengdu, China). The structures of the above eight reference standards are shown in Figure 6. CMC-Na was purchased from Chengdu Kelong Chemical Co., Ltd., (Chengdu, China). Spectro grade methanol, formic acid and acetonitrile were purchased from Thermo Fisher Scientific (Shanghai, China) Co., Ltd. Watson’s distilled water was purchased from Watson’s food and Beverage Guangzhou Co., Ltd., (Guangzhou, China). All other chemicals were of analytical reagent grade.

### 3.2. Animals

Thirty healthy male C57BL/6J mal mice (6–8 weeks, weighing 18–22 g) were purchased from Hunan Slack Jingda Experimental Animal Co., Ltd., (Changsha, China, License No. SCXK, 2019-0004). All mice were housed under standard conditions (room temperature 22 ± 1 °C, humidity 55% ± 5%, 12:12 h light/dark cycle) and were given free access to food and water. Animal care and treatment procedures were approved by the Laboratory Animal Center of Zunyi Medical University (ZMU23-2303-305).

#### 3.2.1. Blood Samples Collection

After acclimating for seven days, all animals were randomly divided into two groups according to body weight. Among them, three mice were assigned to the blank group, and the remaining 27 mice were assigned to the GPs group. The GPs group was orally administered a dose eight times higher than the recommended dose of the 2020 Chinese Pharmacopoeia (216 mg/kg), and the blank group received an equal volume of 0.1% CMC-Na. Blank blood samples were collected from the ophthalmic veins before GPs administration. For the GPs group, blood samples were collected at 10, 30, 45, 60, 120, 240, 360, 480 and 720 min post-administration by anesthetizing three mice at each time point, and blood was collected from the ophthalmic veins (Appendix A). Blood samples were placed at 4 °C for 30 min, centrifuged at 4500 rpm for 15 min, and supernatant was collected. All serum samples were stored at −80 °C until analysis.

#### 3.2.2. Sample Preparation

The serum samples (50 μL) were mixed with three volumes of acetonitrile (150 μL), vortexed for 5 min, and centrifuged at 13,000 rpm for 15 min. The supernatant (150 μL) was evaporated to dryness under nitrogen. The residue was reconstituted in 100 μL of 100% acetonitrile, vortexed for 5 min and centrifuged. Then, the 50 μL supernatant was injected into sampling vials for UPLC-Q-TOF-MS/MS analysis.

### 3.3. UPLC-Q-TOF MS/MS Analysis

A sample analysis was performed on the SHIMADZU LC 40C liquid chromatography system (SHIMADZU, Japan), consisting of an SPD-40V detector, CTO-40C column oven, LC-40B XR solvent delivery unit, SIL-40C autosampler, CBM-40Lite system controller and a Shimadzu LabSolutions workstation. Analytes were separated on an ACQUITY UPLC HSS T3 column (2.1 × 100 mm, 1.8 μm). The mobile phase consisted of 0.1% formic acid water (solvent A) and acetonitrile (solvent B). The following optimized UPLC elution conditions were employed: 0% to 15% B at 0–5 min, 15% to 45% B at 5–10 min, 45% to 70% B at 10–20 min, 70% to 100% B at 20–33 min, and finally maintained at 100% B for 3 min. The column temperature, flow rate and injection volume were 30 °C, 0.3 mL/min and 2 μL, respectively.

Mass spectrometric analysis was performed on an X500R Q-TOF-MS instrument (AB SCIEX, USA) coupled with an electrospray ion source (ESI), and data analysis was conducted using the SCIEX OS (version 3.1.6.44, AB SCIEX, USA) workstation. The operating mass spectrum parameters were as follows: *m/z* 100 to 1500 Da; ion spray voltage, 4.0 kV (ESI+) and −4.5 kV (ESI−); ion source gas 1 (gas 1), 50 psi; ion source gas 2 (gas 2), 50 psi; curtain (CUR) gas, 35 psi; collision associated dissociation (CAD) gas, 8 psi; ion source temperature (Temp), 500 °C. The collision energy (CE) and declustering potential (DP) were set at ±10 V and ±50 V, respectively.

### 3.4. Identification and Characterization of GPs Absorbed Prototypes and Metabolites

#### 3.4.1. Identification of GPs Prototypes

A library for the chemical compound of GPs was established, containing compound names, chemical formula, molecular weight and CAS, sourced from CNKI, PubMed, Web of Science and other databases or platforms (Appendix A). Then, the mass spectral information for the identified compounds in the reference, GPs prototype herbs and the gypenosides component database were combined to identify the prototypic components of GPs in mice serum, with a mass spectral deviation controlled within ±5 ppm.

#### 3.4.2. Characterization of GPs Metabolites

The metabolite identification process was divided into theoretical prediction and manual validation. Theoretically, components entering the body can undergo 71 different metabolic reactions, resulting in the production of 71 metabolites (Appendix A). Consequently, the metabolic reactions occurring in vivo for the saponin components of GP were simulated based on the known compounds in GP using the “R” language, resulting in the generation of a virtual metabolite database (Appendix A). The code for generating the virtual metabolite database was provided in Appendix A. To refine the scope of potential metabolites in GPs and acquire higher-quality MS/MS data, this study also utilized “R” language to define the polygonal mass deficit regions for potential metabolites of GPs by using the integer part and decimal part after expanding each metabolite *m/z* value by a factor of 1000 as the *x*–axis and *y*–axis, respectively. Then, the raw data from UPLC-Q-TOF-MS/MS were subjected to deconvolution, peak alignment and peak extraction using SCIEX OS (version 3.1.6.44) to generate a mass spectrometry data matrix for the mice serum administrated with GPs. Subsequently, the point. in. polygon function, as part of the SP package within the “R” language, was employed to expeditiously screen the mass spectrometry data within the mass spectrometry data matrix, which conformed to the polygonal mass region for potential prototype components and metabolites of GPs. During the screening process, an allowable range of ±5 ppm between mass errors was permitted. The code for generating and using polygon mass defect filtering for screening potential prototypes and metabolites of GPs was provided in Appendix A. Next, based on the nominal mass (NMRU) and exact mass (MRU) of the repeating unit (RU), the mass (M) of each screened compound was converted to the Kendrick mass (KM) using the formula (1). Kendrick mass deficit (KMD) was calculated based on the nominal Kendrick mass (NKM) using formula (2) [39]. This process enabled the rapid characterization of methylated, demethylated (RU = CH_2_), oxidative, and deoxidized (RU = O) metabolites of GPs. The PeakView (version 1.2) was performed for further validation based on precursor ions and metabolite data. The mass spectrum information of the serum sample was compared with the theoretical metabolite data for the 71 predicted metabolites. If the MS and MS^2^ data were consistent with the theoretical metabolite data, the metabolite was considered present in the sample. Otherwise, the predicted metabolite does not exist.(1)KM=M∗NMRU/MRU(2)KMD=NKM−KM

### 3.5. Application the Feasibility of the Integrated Metabolite Annotation Strategy

To further confirm the feasibility of the proposed strategy and validate the characterization of GPs prototype and metabolite, Gyp LXXV, one of the characterized prototypes, was used as an example to verify its absorption and to further characterize its metabolites. The specific operations conducted were as follows: mice serum samples were collected following the administration of Gyp LXXV (100 mg/kg), and the same chromatographic, mass spectrometry, and data analysis methods employed in items “2.4” and “2.5” were utilized to identify Gyp LXXV and its metabolites in mice serum.

## 4. Conclusions

Identifying the prototypic components and metabolites of natural products in vivo remains a challenging task. To quickly and accurately screen and characterize the absorbed prototypes and metabolites of natural products from a large amount of mass spectrometry data, this study developed a rapid and accurate characterization strategy for the prototype components and metabolites of natural products absorbed into the blood by using the R programming language on the basis of the establishment of GPs chemical component database. The proposed strategy was applied to the characterization of prototype components and metabolites in the serum of mice given GPs. As a result, 719 potential metabolites of GPs were screened from mice serum by employing polygonal mass deficit, and 36 prototypic components and 108 metabolites were rapidly characterized in mice serum by employing Kendrick mass deficit. Finally, the strategy was validated using Gyp LXXV as an example, and the results demonstrated that the established method can rapidly identify the metabolites of Gyp LXXV. In addition, most of the metabolites are obtained by phase I metabolic reactions such as hydroxylation and methylation after the hydrolysis of sapogenins, and a few metabolites are phase II metabolites such as acetylation and glucuronidation products of sapogenins. This study not only provides a valuable perspective for the rapid and accurate characterization of prototype components and metabolites of natural products absorbed into the blood but also scientifically elucidates the metabolic pathway of GPs in vivo for the first time.

## Figures and Tables

**Figure 1 molecules-30-00779-f001:**
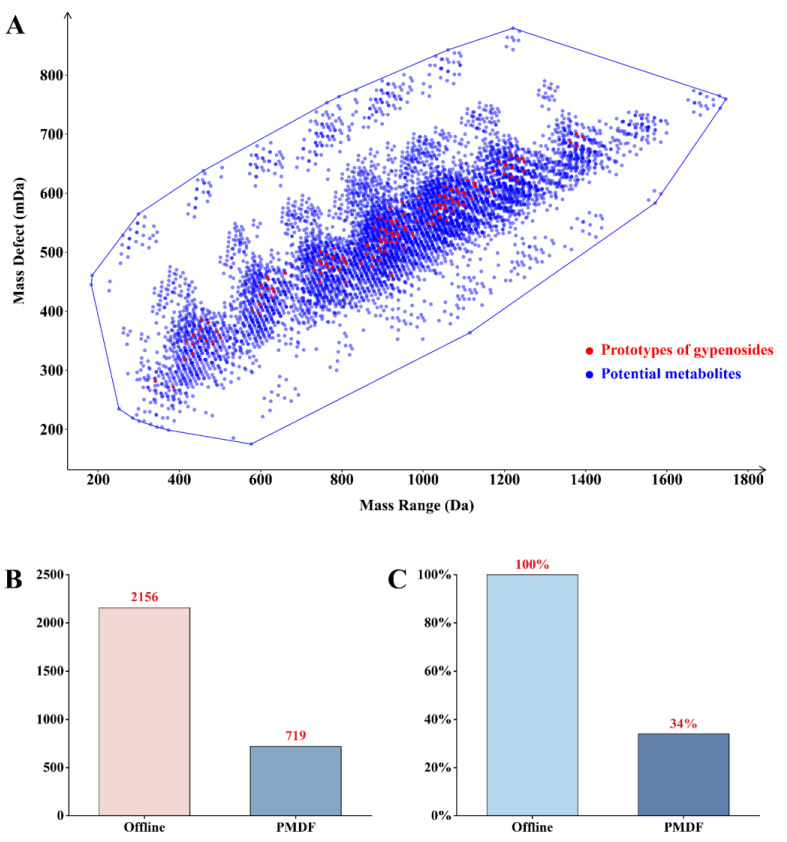
Polygonal mass defect plots for GPs based on the self-built virtual metabolite library and metabolite screening results after administering GPs using the “R” programming language. (**A**) mass defect regions for potential saponin metabolites in GPs; (**B**) screening results (absolute value) of potential saponins and their metabolites based on polygonal mass defect in combination with a virtual metabolic product library; (**C**) screening results (percentage values) of potential saponins and their metabolites based on polygonal mass defect in combination with a virtual metabolic product library.

**Figure 2 molecules-30-00779-f002:**
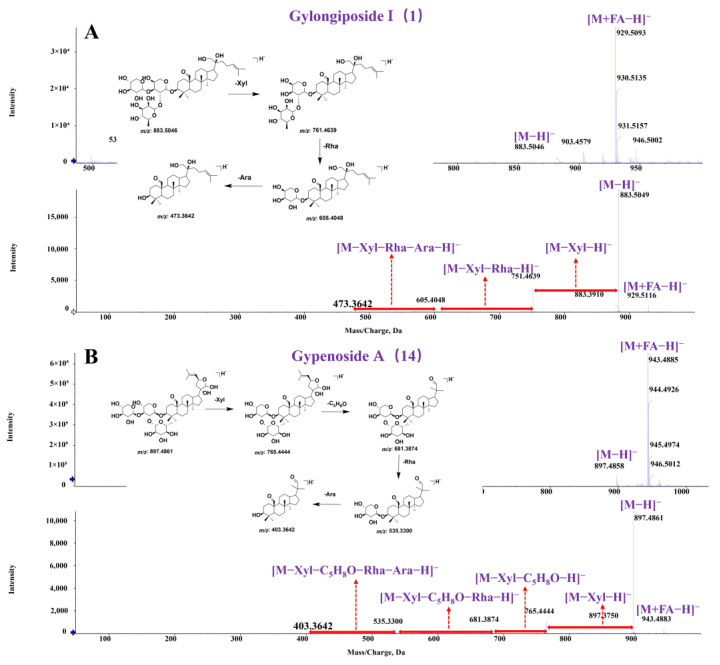
MS and MS^2^ spectra of gypenosides components in mice serum, along with potential mass spectrometry fragmentation patterns. (**A**) MS and MS^2^ spectra and potential fragmentation patterns of gylongiposide I; (**B**) MS and MS^2^ spectra and potential fragmentation patterns of gypenoside A.

**Figure 3 molecules-30-00779-f003:**
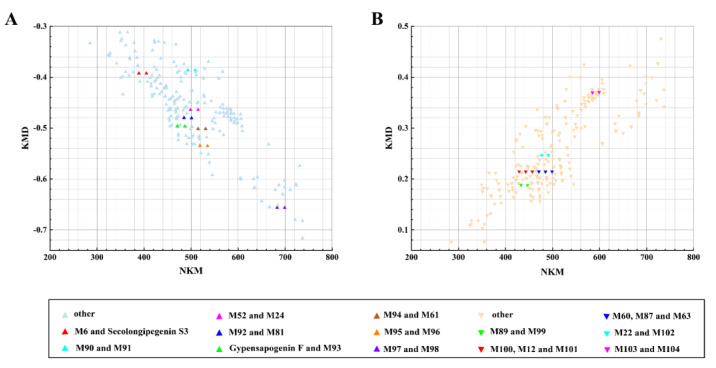
Targeted screening of metabolites of GPs absorbed components in phase I metabolism via deoxidized, oxidation, demethylation and methylation reactions using Kendrick mass defect filtering. (**A**) Screening process of metabolites after oxidation and deoxidized reactions of GPs; (**B**) screening process of metabolites after methylation, demethylation reactions of GPs.

**Figure 4 molecules-30-00779-f004:**
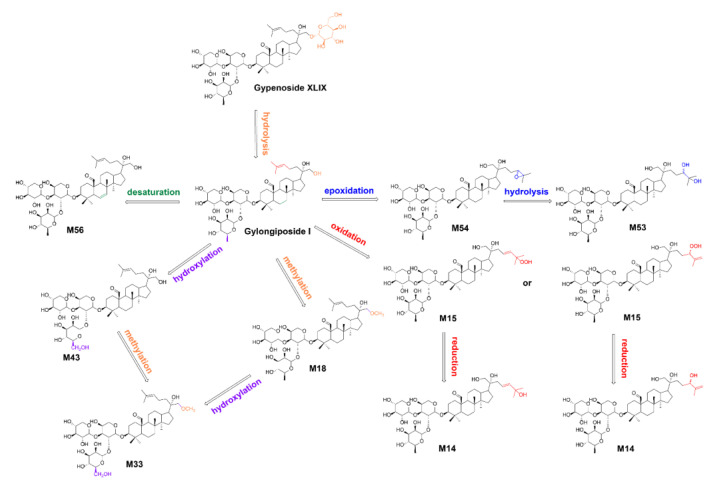
The metabolic pathway of gypenoside XLIX in vivo.

**Figure 5 molecules-30-00779-f005:**
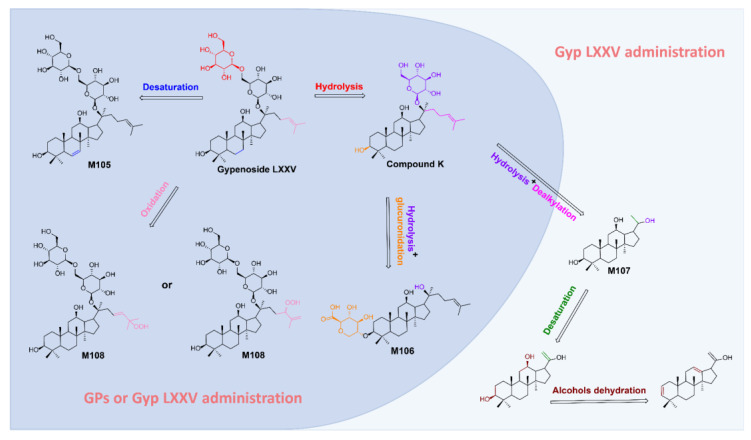
In vivo metabolic pathway of Gyp LXXV in the GPs-administrated and Gyp LXXV-administrated mice.

**Figure 6 molecules-30-00779-f006:**
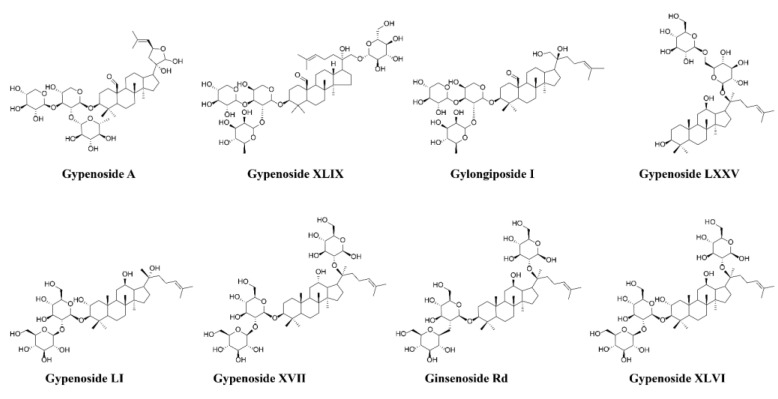
Chemical structures of 8 reference compounds.

## Data Availability

Data is contained within the article.

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
