# Peer review of "Utilizing High-Resolution Mass Spectrometry Data Mining Strategy in R Programming Language for Rapid Annotation of Absorbed Prototypes and Metabolites of Gypenosides"

_molecules, 2025, doi:10.3390/molecules30040779_

Round 1
Reviewer 1 Report
Comments and Suggestions for Authors
This work utilized R to find metabolites of Gypenoside based on mass defect analysis. This method is straightforward while the authors could improve their reproducibility.
I checked their R code and it only include the polygonal mass defect analysis part. The KMD analysis should also be included as figure 6 didn't show the RU for those analysis. It's also recommended to include the peak list to run the analysis.
Section 3.3 is an application instead of validation for this method. Validation need standards to match the proposed annotation.
Figure 2, 3C, 5 can be moved to SI with limited information for the reader.
Caption of figure 3: mass defect plot instead of mass deficit plot
Author Response
Comments 1: [I checked their R code and it only include the polygonal mass defect analysis part. The KMD analysis should also be included as figure 6 didn't show the RU for those analysis. It's also recommended to include the peak list to run the analysis.]
Response 1: [We thank the reviewer for the comment. In this study, the processing of PMDF is the most complicated for annotations of the prototype components and their metabolites in Gypenosides Compared with using R code to process KMD data, it is faster to edit formulas in Excel in Microsoft software. For this reason, the R code of this study is only used in the PMDF. In addition, we have added peak list (table S6 and table S7) for the data used in Figure 6 in the supplementary materials.]
Comments 2: [Section 3.3 is an application instead of validation for this method. Validation need standards to match the proposed annotation.]
Response 2: [We appreciate the suggestions of the reviewers. We have changed the title in section 3.3 to the application of rapid identification methods for natural product metabolites.]
Comments 3: [Figure 2 and 5 can be moved to SI with limited information for the reader.]
Response 3: [We appreciate the suggestions of the reviewers. We have moved Figure 2 and 5 to supplementary materials.]
Comments 4: [Caption of figure 3: mass defect plot instead of mass deficit plot.]
Response 4: [We thank the reviewer for the comment. We have changed the title of Figure 3.]
Reviewer 2 Report
Comments and Suggestions for Authors
This manuscript describes a characterization strategy for the prototype components and metabolites of natural products in blood using the R programming language. The proposed strategy has not only been applied to the characterization of prototype components and metabolites in mouse serum but has also been validated by Gyp LXXV. This manuscript is well written and organized and is helpful in the characterization of prototype components and metabolites of natural products absorbed into the blood. The drawbacks of this work are not focused on the characterization strategy, namely the available features of the strategy, but pays much attention to the metabolic pathway of gypenoside XLIX. If the latter is the main novelty, the topic of this work needs to be revised.
If possible, please compare the developed strategy of using the R programming language to screen and annotate potential prototype components and metabolites with the available ones.
Also, although the language is not a barrier for me to understand this work, there are some grammatical errors that need to be corrected.
The peaks in the mass spectra of Figure 4 are too blurred to be observed.
Comments on the Quality of English LanguageAs the comments and suggestions for authors
Author Response
Comments 1: [This manuscript describes a characterization strategy for the prototype components and metabolites of natural products in blood using the R programming language. The proposed strategy has not only been applied to the characterization of prototype components and metabolites in mouse serum but has also been validated by Gyp LXXV. This manuscript is well written and organized and is helpful in the characterization of prototype components and metabolites of natural products absorbed into the blood. The drawbacks of this work are not focused on the characterization strategy, namely the available features of the strategy, but pays much attention to the metabolic pathway of gypenoside XLIX. If the latter is the main novelty, the topic of this work needs to be revised.]
Response 1: [We thank the reviewer for the comment. We admit that this method is not perfect enough, and we will further improve the method in the later study. However, the method we have developed is universal for prototype components and metabolites of complex systems. Specifically, for other types of ingredients, we can complete the screening of prototype components and metabolites of different types of ingredients by replacing the chemical components in the self-built database.]
Comments 2: [If possible, please compare the developed strategy of using the R programming language to screen and annotate potential prototype components and their metabolites with the available ones.]
Response 2: [We appreciate the suggestions of the reviewers. Annotations of existing natural product prototype components and metabolites are mainly identified by reference, open source and commercial databases. With the improvement of the mass spectrometer, the acquisition speed and sensitivity are continuously improved, and the data collected is becoming larger and larger. In this way, there will be more potential prototype components and metabolites screened by traditional annotation methods, and it will take a lot of time to further determine. Therefore, the use of R language to screen and annotate prototype components and their metabolites is mainly faster and more accurate than the existing methods. We have added corresponding comparative information to the manuscript.]
Comments 3: [Also, although the language is not a barrier for me to understand this work, there are some grammatical errors that need to be corrected.]
Response 3: [ We appreciate the suggestions of the reviewers. We have carried out a comprehensive review of the full text of the English text, and the problematic areas have been are shown in blue text.]
Comments 4: [The peaks in the mass spectra of Figure 4 are too blurred to be observed.]
Response 4: [We appreciate the suggestions of the reviewers. We have modified the mass spectrum peak in Figure 4.]